# Repeat DNA-PAINT suppresses background and non-specific signals in optical nanoscopy

Alexander H. Clowsley [1], William T. Kaufhold[2,3], Tobias Lutz[1], Anna Meletiou [1], Lorenzo Di Michele [3] & Christian Soeller [1✉]

DNA-PAINT is a versatile optical super-resolution technique relying on the transient binding of fluorescent DNA 'imagers' to target epitopes. Its performance in biological samples is often constrained by strong background signals and non-specific binding events, both exacerbated by high imager concentrations. Here we describe Repeat DNA-PAINT, a method that enables a substantial reduction in imager concentration, thus suppressing spurious signals. Additionally, Repeat DNA-PAINT reduces photoinduced target-site loss and can accelerate sampling, all without affecting spatial resolution.

[1] Living Systems Institute and Biomedical Physics, University of Exeter, Exeter EX4 4PY, UK. [2] Cavendish Laboratory, University of Cambridge, Cambridge CB3 0HE, UK. [3] Department of Chemistry, Molecular Sciences Research Hub, Imperial College London, London W12 0BZ, UK. ✉email: C.Soeller@exeter.ac.uk

Super-resolution optical microscopy methods have become essential tools in biology, and among these DNA-PAINT[1–5] has proved especially versatile[6,7]. In DNA-PAINT, epitopes of interest are labeled with 'docking' DNA motifs, while dye-modified 'imager' oligonucleotides are introduced in solution. Transient hybridization to docking motifs immobilizes imagers for long enough to generate 'blinks' (events) in a camera frame, which can then be fitted to localize target epitopes with sub-diffraction resolution[2]. DNA-PAINT carries several advantages compared to competing approaches such as STORM[8,9] and PALM[10,11], eliminating the need for photo-switchable or chemically-switchable dyes and effectively circumventing photo-bleaching, due to fresh imagers continuously diffusing in from the bulk.

The unparalleled flexibility of DNA-PAINT comes at a cost, in the form of a number of serious drawbacks currently limiting the applicability and performance of the technology when imaging biological cells and tissues.

The presence of free imagers in solution produces a diffuse fluorescent background, which compromises event detection and localization precision. The impact of free-imager signals is particularly severe when imaging deep in biological tissues, where efficient background-rejection methods such as TIRF cannot be used.

In addition, imagers often exhibit substantial non-specific binding to biological preparations, which complicates data interpretation[7] and can prevent detection of sparse targets[12].

Both imager-induced background and non-specific events can be reduced by decreasing imager concentration. However, such a reduction also decreases event rates and extends image-acquisition timescales, which is often prohibitive due to limitations in mechanical and chemical sample stability.

Finally, despite it being effectively immune to photobleaching, DNA-PAINT has been shown to suffer from photo-induced inactivation of docking strands[13].

Here, we introduce repeat DNA-PAINT, a straightforward strategy that mitigates all these critical limitations of DNA-PAINT.

## Results

**Repeat DNA-PAINT affords an increase in event rate.** As demonstrated in Fig. 1a, c, we employ docking motifs featuring $N$ identical Repeated Domains ($N$x RD, $N = 1, 3, 6, 10$) complementary to imagers. Unless otherwise specified, we use a 9-nucleotide (nt) imager (P1) whose concentration is referred to as $[I]$.

In the super-resolution imaging regime, only a small fraction of docking sites is occupied by imagers at any given time. In these conditions, and if all repeated docking domains are equally accessible to imagers as in a 1x RD motif, the spatial event density $E$ is expected to be proportional to the product of imager concentration and repeat domain number $N$:

$$E = \rho_{DS} \cdot N \cdot \frac{[I]}{K_d}, \quad (1)$$

where $\rho_{DS}$ is the docking strand density (set by the density of markers in the sample) and $K_d$ the binding affinity of imagers to a single docking domain (see also Supplementary Note 1).

In agreement with Eq. 1, tests performed on functionalized microspheres demonstrate a linear growth in event rate with increasing $N$, for fixed imager concentration $[I] = 50$ pM (Fig. 1b). The experimental findings are confirmed by molecular simulations, relying on the oxDNA[14] model and the Forward–Flux Sampling method to estimate imager-docking binding rates[15] (Fig. 1b).

Simulations further highlight that, as expected, imagers bind all individual domains on the repeat-docking motifs with similar probability, proving that the elongation of docking motifs does not hinder their accessibility (Supplementary Fig. 1).

Equation 1 also indicates that, when using docking motifs with $N$ repeats, the imager concentration can be reduced $N$-fold while preserving the event density $E$, or equivalently the event rate (when summed over a region of interest and quantified per frame).

To confirm this hypothesis we constructed DNA origami test tiles that display a number of "anchor" overhangs, initially connected to 1x RD docking motifs. The former could then be displaced through a toe-holding reaction, and were replaced with a 10x RD strand (Fig. 1c). The event rate per origami tile was preserved when changing from 1x RD docking sites with 0.4 nM imager concentration to 10x RD docking sites but 10-times lower imager concentration of 40 pM (Fig. 1d). The same strategy was applied to biological samples, specifically cardiac tissues[6] where we labeled ryanodine receptors (RyRs) with the common anchor strand that initially held a 1x RD motif. As expected, we find near identical event rates when imaging 1x RD with $[I] = 0.4$ nM versus replacing these with 10x RD with $[I] = 40$ pM (Supplementary Fig. 2).

**Repeat DNA-PAINT suppresses backgrounds and enhances resolution.** The ability of Repeat DNA-PAINT to function optimally with a substantial (up to 10-fold) reduction in imager

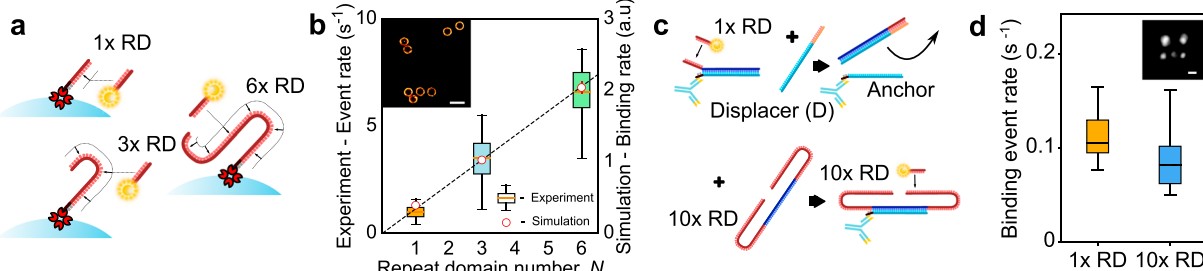

**Fig. 1 Repeat DNA-PAINT preserves event rates at greatly reduced imager concentration. a** Docking motifs with $N = 1, 3$, or 6 binding sites, here biotin-modified and anchored to streptavidin-coated microspheres. **b** The event rate scales linearly with $N$, as determined experimentally on microsphere test samples and by coarse-grained computer simulations. The dashed line is a linear fit to the simulation results. Inset: rendered image of a selection from one set of functionalized microspheres. $n = 82, 88, 68$ microspheres for 1x, 3x, 6x RD, respectively. **c** Scheme enabling swapping between 1x RD and 10x RD docking motifs. A common anchor strand is first connected to a 1x RD strand, which can be removed with a displacer strand D and replaced with a 10x RD motif. **d**: Application of the scheme in **c** on synthetic origami tiles shows the number of events per second per tile remains approximately the same with origami functionalized with either 1x or 10x RD when using nominally 10-fold decrease in imager concentration. $n = 49$ origami tiles (1x RD) and $n = 81$ origami tiles (10x RD). Boxplots show minima, maxima and median of the data. Scale bars: **b** 1 μm, **d** 30 nm.

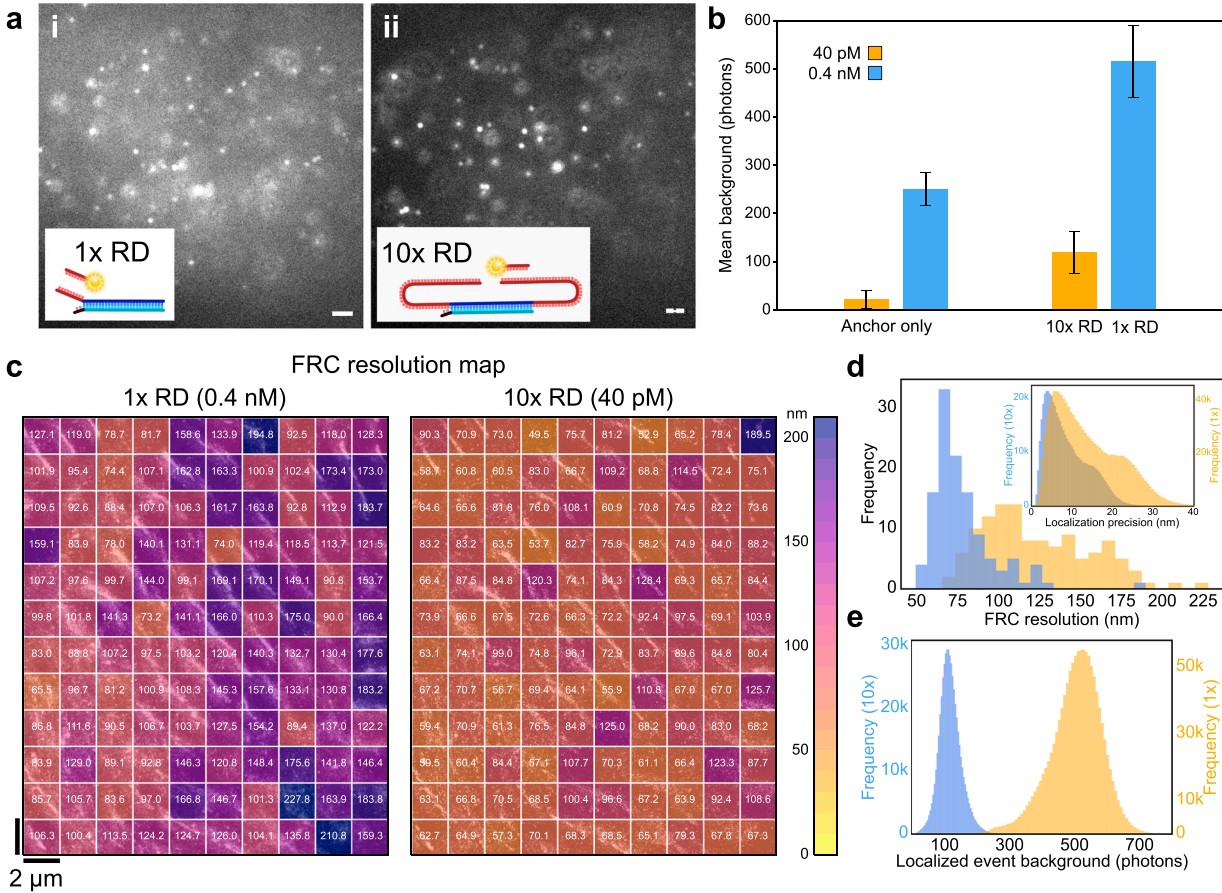

**Fig. 2 Repeat DNA-PAINT allows reducing backgrounds and increases image resolution. a** Raw camera frames of cardiac tissue labeled for RyRs and recorded using 1x RD with $[I] = 0.4$ nM (**i**), and then 10x RD with $[I] = 40$ pM (**ii**), using the scheme in Fig.1c. Note the lower background and better contrast in ii. Similar levels observed in $n = 5$ repetitions in biological tissue. **b** Background increases approximately proportionally with imager concentrations in tissue containing only anchor strands (n = 712 events for $[I] = 40$ pM, 5002 events for $[I] = 0.4$ nM), i.e. without complementary binding sites. Once functionalized with 1x RD ($n = 683$k events) or 10x RD ($n = 537$k events), out-of-focus binding events contribute to an additional offset. P1 imager concentration 40 pM (orange) and 0.4 nM (blue). Event frequencies were similar in both modalities (1x–10x RD) with an order of magnitude difference in imager concentration, comparable to other biological experiments (Supplementary Fig. 2). Error bars are SD around the mean. **c** Fourier Ring Correlation resolution maps, displaying the resolution in nm per segment, calculated for 1x RD and 10x RD imaging runs of the same region in a thick (20 µm) tissue section labeled for alpha actinin, see also Supplementary Fig. 3. **d** The improvement in FRC resolution, taken explicitly from the FRC resolution maps in **c**, can be attributed to improved localization precision (see inset) which results from the reduced background present in the 10x RD data (**e**), due to the lower imager concentration. Scale bars: **a** 2 µm.

concentration makes it ideal for mitigating issues resulting from imagers in solution, the most direct being the fluorescent background produced by unbound imagers.

In Fig. 2 we therefore investigate the fluorescent background in cardiac tissue samples with conventional docking strands (1x RD) and repeat domains (10x RD). Visual assessment demonstrates a clear improvement in contrast between the two imaging modes, as shown by example frames in Fig. 2ai (1 RD) and Fig. 2 aii (10x RD), to an extent that substantially improves the detectability of individual binding events and their localization precision[16].

For a quantitative assessment, we measured background signals produced with $[I] = 40$ pM and 0.4 nM in optically thick tissues labeled with common anchor overhangs, but lacking docking motifs. Figure 2b (left pair of bars), demonstrates a near linear increase of the fluorescent background with $[I]$. Once the markers were functionalized with docking strands, either 1x RD or 10x RD, the ratio of background levels was slightly lower, apparently due to an additional offset background (Fig. 2b, right pair of bars). We hypothesize that the additional background is generated by specific binding events occurring out of the plane

of focus. These events are indeed expected to produce an out-of-focus signal proportional to the event rate, and thus similar when using 1x RD with 0.4 nM versus 10x RD with 40 pM of imager (by design).

It is expected that the substantial reduction in background afforded by Repeat DNA-PAINT translates into a significant improvement in resolution. To quantify this improvement we imaged deep (several microns) into optically thick (~20 µm) cardiac tissue using this technique. We performed a two-stage experiment as exemplified in Fig. 1c, first imaging with 1x RD at high $[I]$ and then with 10x RD at low $[I]$. In both cases, we carried out Fourier Ring Correlation (FRC) measurements of the optical resolution in $2 \times 2$ µm² regions across the ~$24 \times 20$ µm² imaging region (Fig. 2c). This yielded a mean FRC resolution measurement (Fig. 2d) of $123.7 \pm 3.0$ nm (SEM) for 1x RD, $[I] = 0.4$ nM, and $78.0 \pm 1.8$ nm (SEM) for 10x RD, $[I] = 40$ pM, confirming the substantial improvement in resolution with Repeat DNA-PAINT when background from imagers in solution cannot be effectively rejected, e.g., when imaging deep in thick tissue with widefield illumination (Fig. 2e).

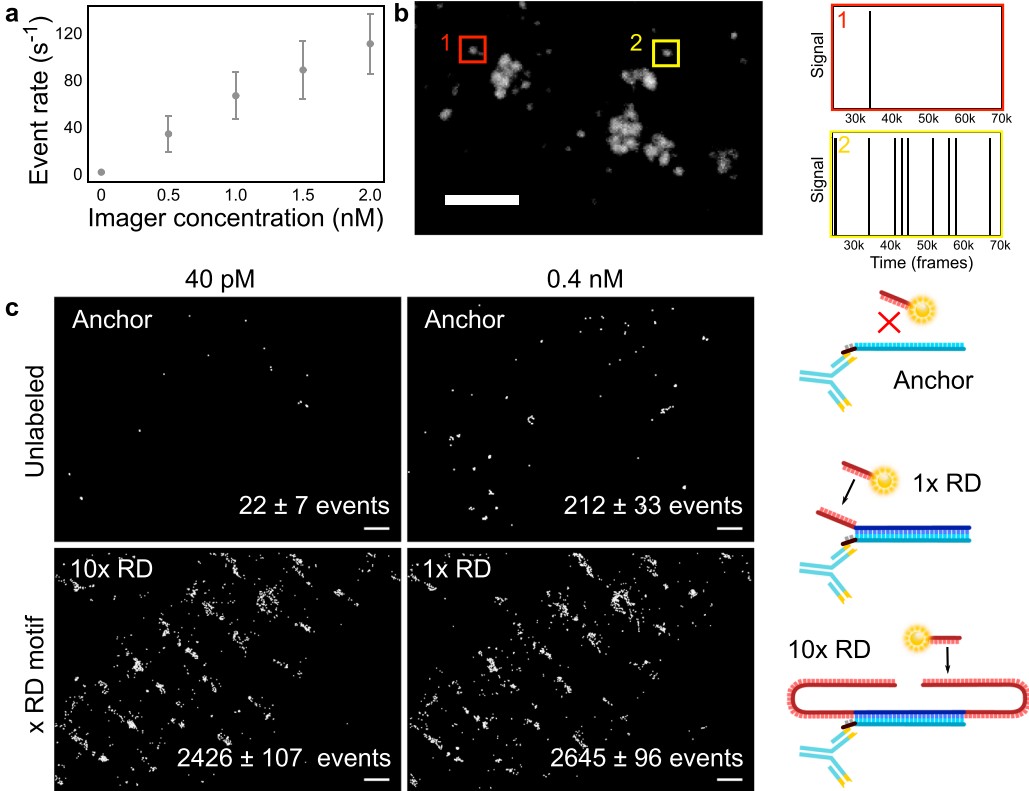

**Fig. 3 Repeat DNA-PAINT reduces non-specific imager binding. a**: Rate of non-specific binding events of P1 imagers in unlabeled cardiac tissue as a function of [*I*], displaying a linear trend. Error bars show the SD around the mean of the number of binding events per second, ($n = 200$). **b**: DNA-PAINT image of RyRs obtained using the protocol in Fig. 1c and rendering only the 1x RD events. Time traces of typical 1x RD showing suspected non-specific (1) and specific events (2) from the regions highlighted. Similar event kinetics observed across $n = 5$ repetitions in biological tissue. **c**: A visual example of the levels of non-specific binding present in a typical experiment. First, tissue was imaged with [*I*] = 40 pM and 0.4 nM when only anchor sites were present in order to obtain the rates of non-specific events. Once the anchor strands were functionalized with 1x RD (and [*I*] = 0.4 nM) or 10x RD (and [*I*] = 40 pM), event rates are comparable, suggesting that 1x RD signals contain ~8% non-specific events whilst 10x RD signals have only ~0.9% contribution from non-specific events. Similar levels of non-specific events observed in $n = 3$ comparable repetitions. Scale bars: **b** 500 nm, **c** 1 µm.

**Repeat DNA-PAINT suppresses non-specific binding**. Having proven the benefits of Repeat DNA-PAINT in reducing backgrounds and improving resolution, we assessed its impact on non-specific imager-binding events at unlabeled locations of biological samples. These non-specific events produce spurious blinks that are often difficult to distinguish from proximal specific signals. Expectedly, Fig. 3a shows that the rate of non-specific events, as detected in unlabeled cardiac tissue, scales linearly with [*I*]. Similar trends are observed for different imager sequences (Supplementary Fig. 4).

In Fig. 3b we study the time-sequence of imager-attachment events recorded in cardiac tissue, as a potential way of separating specific from suspected non-specific events. We compare a trace recorded within a likely unlabeled area, where only suspected non-specific events are observed, based on only one brief attachment phase (Fig. 3b, red region), with one measured at a location where docking strands are present and specific binding is detected (Fig. 3b, yellow region). We observe a qualitative difference between the two situations, with specific binding occurring steadily and suspected non-specific events being often localized in time[1], similar to the time courses of imager attachment observed in data from unlabeled cardiac tissue, which underlies the summary data in Fig. 3a.

Although occasionally applicable, this identification strategy is only robust if specific and suspected non-specific binding sites are spatially isolated. In samples where docking strands are more densely packed and/or evenly distributed, non-specific events cannot be easily separated (Supplementary Fig. 5), introducing potential artifacts in the reconstructed images and distorting site-counting as performed, e.g., via qPAINT[3].

Repeat DNA-PAINT offers a solution that avoids the complexity of identifying non-specific events, by directly reducing their occurrence to negligible levels, as demonstrated in Fig. 3c. Specifically, owing to the 10-fold reduction in imager concentration, image data collected with 10x RD on our cardiac samples only feature ~0.9% non-specific events, whereas conventional DNA-PAINT, here implemented with 1x RD docking strands, yields a ~8% non-specific contamination. We thus conclude that Repeat DNA-PAINT offers a robust route for suppressing spurious events independent of sample characteristics.

**Repeat DNA-PAINT mitigates photoinduced site damage**. Despite its insensitivity to photobleaching, DNA-PAINT is subject to a progressive inactivation of docking sites, ascribed to their interaction with the free-radical states of photo-excited fluorochromes[13]. The domain redundancy in Repeat DNA-PAINT can greatly slow down site loss, as we demonstrate with origami test tiles nominally featuring six anchor sites (Fig. 4a). For tiles with 1x RD and 10x RD motifs, we compare the average number of sites actually detected on the tiles in the first 20 K frames of long imaging runs, to those counted in the following 20 K frames. While for 1x RD tiles we observed a ~12.1% loss of docking sites between the two experimental intervals, 10x RD tiles just lose

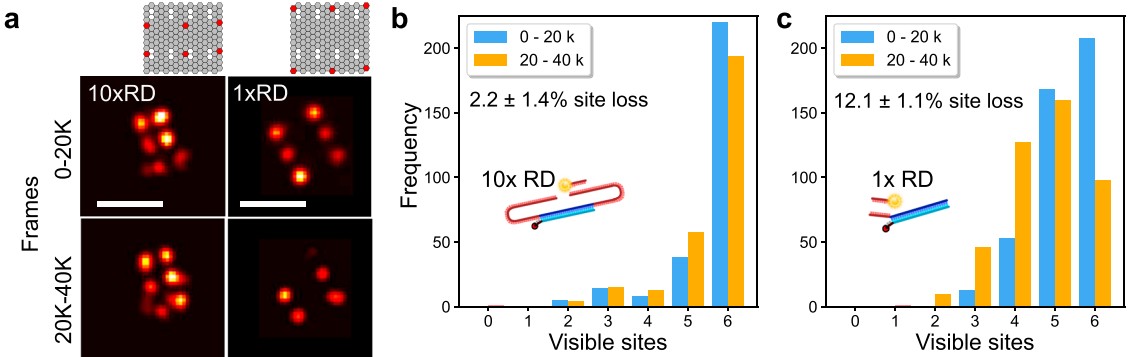

**Fig. 4 Repeat DNA-PAINT reduces docking site-loss. a** Photoinduced site loss as quantified with DNA origami tiles labeled with 10x RD or 1x RD by comparing the number of sites detected in the first half (0–20 K frames) versus the second half (20–40 K frames) of an experimental run. (left) Rendered images of typical tiles, origami designs as shown at top. (right) Histograms summarizing the percentage of lost sites, using (**b**) 10x RD or (**c**) 1x RD. Site loss is much more extensive when using 1x RD docking strands. Scale bars: 100 nm.

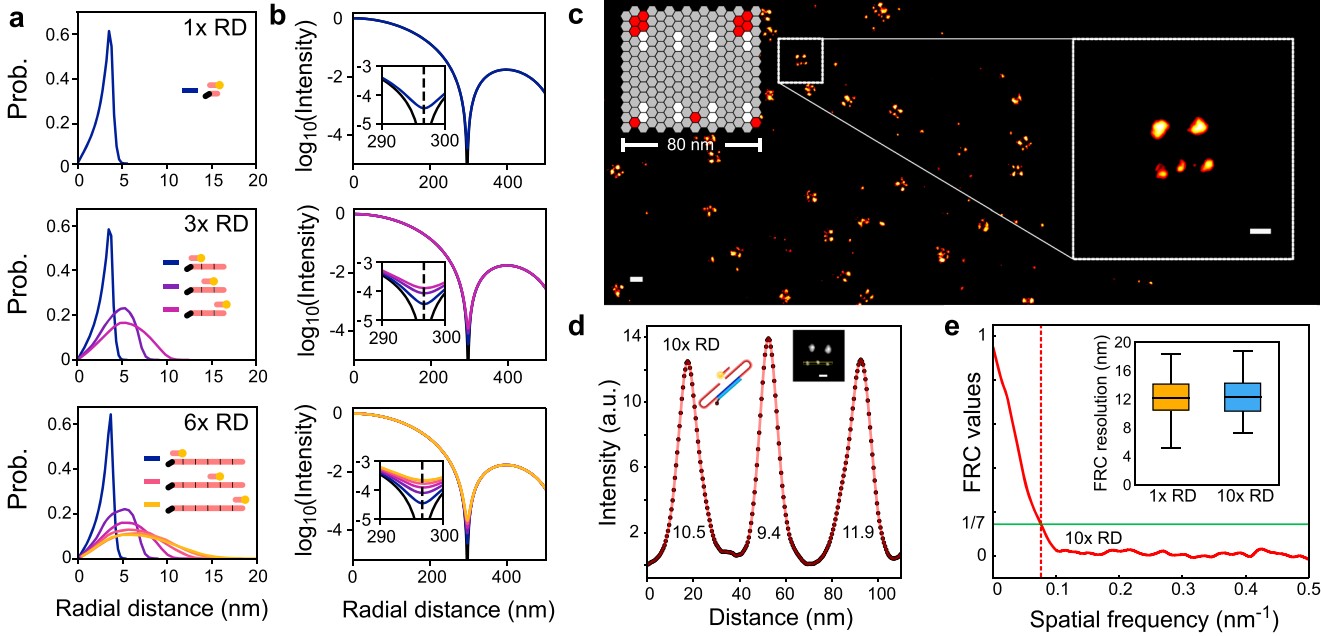

**Fig. 5 Repeat DNA-PAINT preserves spatial resolution. a** Simulated radial distributions of the fluorophore site on imagers hybridized to all possible sites on 1x RD, 3x RD, and 6x RD, with respect to the anchoring point of the docking motif. **b** Radial profiles of blinks as obtained by convolving the fluorophore-distributions with the microscope point-spread function (Supplementary Fig. 6). Insets: zoom of the region around the first Airy minimum, showing very small broadening that is unlikely to be experimentally detectable. **c** Scheme of DNA-origami test tiles with red sites indicating the locations of 10x RD motifs and a rendered DNA-PAINT image, similar origami quality observed in $n = 7$ origami experiments. **d** Typical spatial profiles measured across the 'spots' of origami tiles with 10x RD strands as in **c**, with full-width at half maximum (FWHM) spot diameters as indicated. The average FWHM is 12.28 ± 1.77 nm (mean ± SD), nearly identical to 12.56 ± 2.09 nm determined for 1x RD (Supplementary Fig. 7). **e** Fourier Ring Correlation (FRC) resolution measurements of DNA-PAINT images of origami tiles with 1x RD strands (12.12 ± 2.69 nm, mean ± SD) are indistinguishable from 10x RD (12.36 ± 2.67 nm). Boxplots show minima, maxima and median of the data. Scale bars: 30 nm.

~2.2% (Fig. 4b, c), a 5-fold suppression. Direct examination of the histograms describing the distribution of detectable sites per tile show that with 1x RD more than 50% of the initially complete tiles lost at least one site (Fig. 4b). In turn, the vast majority of complete DNA origami tiles remained intact when using 10x RD docking strands (Fig. 4c).

**Extended docking motifs do not affect spatial resolution.** A potential issue deriving from the extension of the docking strands is the loss of spatial resolution[17,18], as the flexible docking-imager complexes undergo rapid thermal fluctuations during binding events (see Supplementary Note 2). We used oxDNA simulations to quantify the resulting 'blurring', by sampling the distance

between the tethering point of the docking strand and the fluorophore location of imagers hybridized to each binding site in 1x RD, 3x RD, and 6x RD motifs. The results, summarized in Fig. 5a, demonstrate narrow fluorophore distributions for the binding sites closest to the tethering point, and broader ones for the more distal sites, peaking at ~8 nm for the furthest domain.

Although this level of broadening may appear significant compared to the resolution of DNA-PAINT in optimal conditions (~5 nm[19]), it has little impact on the precision with which one can localize the labeled epitope by fitting the diffraction-limited image of a blink. The effect can be quantified by convolving the fluorophore distributions (Supplementary Fig. 6 and Supplementary Note 2) with the theoretical point-spread

function (PSF) of the microscope, as shown in Fig. 5b. The PSF broadening is minute and produces, at most, a 0.12% shift in the location of the first Airy minimum.

We thus do not expect that the larger physical size of multi-repeat docking motifs cause any loss of experimental resolution. We confirmed this prediction with DNA-origami test samples (Fig. 5c), showing no detectable resolution difference between 1x RD and 10x RD, both rendering spots with apparent diameter of ~13 nm (Fig. 5d and Supplementary Fig. 7). Similarly, the Fourier Ring Correlation (FRC) measure of resolution[20] was essentially unaltered between 1x RD (12.2 ± 2.7 nm) and 10x RD (12.4 ± 2.7 nm) images, as shown in Fig. 5e. Note that when imaging origami test samples, the resolution is virtually unaffected by the higher imager concentration used with 1x RD and the consequent stronger free-imager background, as instead demonstrated for the case of thick biological tissues (Fig. 2). Indeed, origami represent a highly ideal scenario in which imaging can be carried out in TIRF mode, which is highly effective in rejecting out-of-focus backgrounds. Other imaging modes, necessary to investigate thicker biological samples, do not perform nearly as well, leading to the substantial benefits in terms of background and resolution associated with reducing imager concentration.

**Additional advantages of Repeat DNA-PAINT: qPAINT, enhanced imaging rate and photobleaching-free wide-field imaging.** Repeat DNA-PAINT is also fully compatible with extensions of DNA-PAINT, such as qPAINT, a technique that estimates the number of available docking sites within a region of interest. We confirm the accuracy of qPAINT with origami tiles displaying five 10x RD motifs, where the technique estimates 4.93 ± 0.16 sites/tile (see Fig. 6a, and "Methods" section).

In addition, we point out that the boost in event-rate afforded by Repeat DNA-PAINT can also be exploited to increase image acquisition rate.

The key for increasing imaging frame rate is using weakly binding imagers, which thanks to a larger $K_d$, and associated larger off-rate, produce shorter events. In parallel, however, one would have to increase imager concentration in direct proportion to $K_d$, in order to retain a sufficiently high binding frequency, see also Eq. 1 and Supplementary Note 1. The concomitant increase in background (see also Fig. 2) would normally be prohibitive but the event-rate acceleration afforded by Repeat DNA-PAINT allows imaging to be carried out at "normal" imager

concentrations, in the sub nanomolar range. Figure 6b indeed demonstrates that by simply replacing 1x RD with 10x RD at 'conventional' imager concentration ($[I] = 0.3$ nM), and using a shorter (low-affinity) 8 nt imager P1s, one can increase frame rate 10-fold (from 100 ms to 10 ms), and reduce the overall imaging time ~6-fold. When performing accelerated imaging, we observe a slightly lowered limiting spatial resolution, from ~80 nm at 100 ms acquisition time to ~100 nm at 10 ms, see Supplemental Fig. 7. Note however that high frame rate acquisition can be further improved by optimizing illumination conditions, so that the number of photons collected from a dye molecule in a short-exposure frame equals that achieved at longer integration time. The ability of repeated-docking motifs to accelerate imaging has been recently confirmed by Straus et al.[21], which however do not discuss the associated improvements in terms of background, resolution and non-specific signals.

Finally, Repeat DNA-PAINT enables effectively photo-bleaching resistant, high-contrast, diffraction-limited imaging. In all the super-resolution applications described above, low imager concentrations are used so that only a small fraction of docking sites is occupied at any given instant. At higher imaging concentrations, a significant fraction of the sites are occupied by imagers. Since imagers are still constantly exchanged with the surrounding solution, operating under these conditions would in principle allow for photobleaching-free diffraction-limited fluor-escence imaging, including wide-field and point-scanning con-focal. However, to achieve a sufficient docking-site occupancy with conventional 1x RD docking strands, one would have to increase imager concentration to a point where the free-imager background massively reduces contrast. Repeat DNA-PAINT performed with 10x RD motifs solves this issue thanks to the intrinsically higher imager binding rates, which enables wide-field imaging at the imager concentrations normally used for conventional DNA-PAINT. This translates in a straightforward strategy for collecting high-contrast, photobleaching-free images of staining patterns (Supplementary Fig. 9).

## Discussion

In summary, we demonstrate that Repeat DNA-PAINT mitigates all key limitations of DNA-PAINT, namely non-specific events (10x reduction), free-imager background (~5x reduction) and photoinduced site loss (5x reduction) while also being able to accelerate data acquisition (6–10x). We also show that there is no

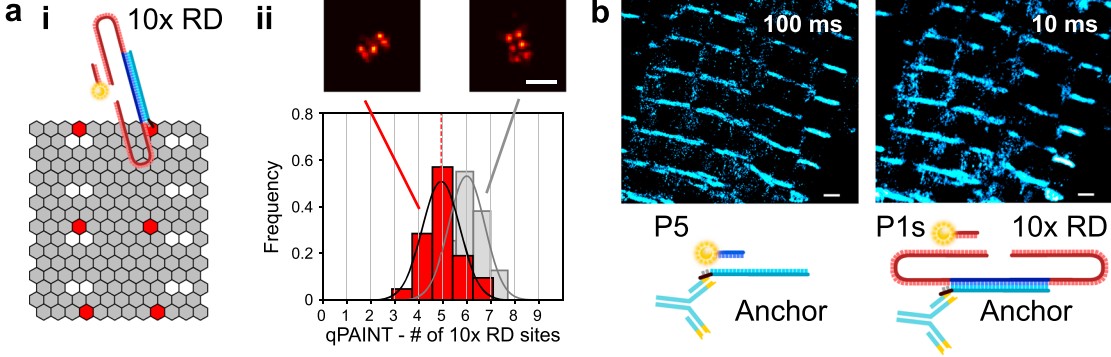

**Fig. 6 Repeat DNA-PAINT is compatible with qPAINT and increases imaging rate. a**i Scheme of the origami test tile used for qPAINT experiments. Due to natural self-assembly inaccuracy, not all tiles feature 6 detectable docking sites. ii Distribution of the number of docking sites determined from qPAINT in tiles featuring 5 detectable sites (red). The median of the histogram is 4.93 ± 0.16 sites/tile. The gray histogram indicates qPAINT results for tiles with 6 sites, used for calibration. Above: rendered images of representative tiles. **b** Rendered DNA-PAINT images of alpha actinin in cardiac tissue as imaged with regular DNA-PAINT (1x RD) at low frame-rate (100 ms/frame, left), and Repeat DNA-PAINT (10x RD) at high frame-rate (10 ms/frame, right), showing similar results. The overall image acquisition time was 2000 s for 1x RD and 1600 s for 10x RD (Supplementary Fig. 8). Samples with 1x RD were imaged with 9 nt P5 imagers. Shorter (8 nt) imagers were used with 10x RD to achieve brief binding times and avoid spatiotemporal overlap of the blinks (see also Supplementary Note 1). In both cases, we used [I] ~0.3 nM. Scale bars: **a** 100 nm, **b** 1 μm.

observable impact on spatial resolution from "long" docking strands containing many repeat domains which greatly extends the design space of Repeat DNA-PAINT. Notably, the implementation of Repeat DNA-PAINT is straightforward and does not carry any known drawbacks, it is routinely applicable, consolidating the role of DNA-PAINT as one of the most robust and versatile SMLM methods.

## Methods

### Experimental methods and materials

*DNA-PAINT oligonucleotides.* Oligonucleotide sequences were designed and checked with the NUPACK web application[22] (www.nupack.org). Oligonucleotides were then purchased from either Integrated DNA Technologies (IDT, Belgium) or Eurofins Genomics (Eurofins, Germany) with HPLC purification. See Supplementary Table 1 for a full list of oligonucleotide sequences used.

*DNA origami production and sample preparation.* All oligonucleotides (staples) used to construct the origami tiles were purchased from IDT with standard desalting, pre-reconstituted in Tris EDTA (10 mM Tris + 1 mM EDTA, TE) buffer (pH 8.0) at 100 μM concentration. Rothemund Rectangular Origami (RRO) with various 3′ overhangs were manufactured following standard methods[2]. Picasso[2] was used to generate staple sequences which yield an RRO with 3′ overhangs in specified locations on a single face of the planar origami. We designed overhangs which would then hybridize to 1x RD or 10x RD docking motifs (see anchor in Supplementary Table 1). Eight DNA strands had 5′ biotin modifications on the reverse face for anchoring. RROs were prepared by mixing in TE + 12.5 mM MgCl₂ the scaffold (M13mp18, New England Biolabs, USA) at a concentration of 10 nM, biotinylated staples at 10 nM, staples featuring the "anchor" 3′ overhangs at 1 μM, and all other staples at 100 nM. Assembly was enabled through thermal annealing (Techne, TC-512 thermocycler) bringing the mixture to 80 °C and cooling gradually from 60 °C to 4 °C over the course of 3 h. A full list of staple sequences can be found in Supplementary Tables 5–7.

Number 1.5 coverslips were submerged in acetone before being moved to isopropanol and subsequently allowed to dry. These were then attached to open-top Perspex imaging chambers as depicted in[23], allowing for easy access. For origami attachment, a 1 mg ml⁻¹ PBS solution of biotin-labeled bovine serum albumin (A8549, Sigma) was applied to the chambers for 5 min and then washed with excess PBS. This was followed by a 1 mg ml⁻¹ solution of NeutrAvidin (31000, ThermoFisher) for a further 5 min before being washed with PBS + 10 mM MgCl (immobilization buffer, IB). DNA-origami solutions were diluted to roughly 1 nM in IB solution and incubated for 5 min on the prepared coverslips. Unbound origami tiles were washed off using excess IB buffer. 1x RD or 10x RD docking motifs were introduced at ~200 nM binding directly to the anchor overhangs on the origami tiles. The samples were then washed with a DNA-PAINT buffer (PB) of PBS containing 600 mM NaCl and pH corrected to 8.0 (adapted from 'Buffer C' in ref.[1]).

*Microsphere functionalization and sample preparation.* Streptavidin-functionalized polystyrene particles with a diameter of 500 nm (Microparticles GmbH, Germany) were labeled with biotinylated oligonucleotides (Fig. 1a: docking motifs 1x RD, 3x RD, and 6x RD, see Supplementary Table 1) as described elsewhere[24]. Briefly the microspheres were dispersed in TE buffer containing 300 mM NaCl and the docking strands in 4x excess concentration as compared to the binding capacity of the beads. Unbound oligonucleotides were removed by a series of centrifugation and re-dispersion steps. These microspheres were attached via non-specific adhesion to coverslips cleaned as described above and coated by incubating them for 30 min with a 0.1 mg ml⁻¹ solution of PLL-g-PEG (SuSoS, Duebendorf) in PBS.

*Oligonucleotide to antibody conjugation.* Anchor oligonucleotides (Supplementary Table 1) were conjugated to secondary antibodies for immunolabelling of cardiac samples. Lyophilized oligonucleotides were resuspended in PBS (pH 7.4) to 100 μM and kept at −20 °C for long term storage until required for conjugation. AffiniPure Goat Anti-Mouse secondary antibodies (affinity purified, #115-005-003, Jackson ImmunoResearch, PA) were conjugated using click-chemistry as described by Schnitzbauer et al.[2] Briefly, the antibody was incubated with 10-fold molar excess DBCO-sulfo-NHS-ester (Jenabioscience, Germany) for 45 min. The reaction was quenched with 80 mM Tris-HCl (pH 8.0) for 10 min and then desalted using 7 K MWCO Zeba desalting columns (Thermo Fisher). A 10-fold molar excess of the azide modified oligonucleotide was then incubated with the DBCO-antibody mixture overnight at 4 °C. Subsequently the antibody was purified using 100 K Amicon spin columns (Sigma). The absorbance of the oligonucleotide-conjugated fluorophores (Cy3 or Cy5) was recorded with a Nanodrop spectrophotometer (Thermo Fisher Scientific, Waltham) and used to quantify the degree of labeling for each conjugation, typically achieving >1–3 oligonucleotides per antibody.

*Biological sample preparation and labeling.* Cardiac tissue (porcine) was fixed with 2% paraformaldehyde (PFA, pH 7.4, Sigma) for 1 h at 4 °C. Samples were then washed in PBS and kept in PBS containing 10% sucrose for 1 h before being moved to 20% (1 h) and finally 30% sucrose overnight. The tissue was then frozen in cryotubes floating in 2-Methylbutane cooled by liquid nitrogen for 10–15 min. Pre-cleaned number 1.5 glass coverslips were coated for 15 min using 0.05% poly-L-lysine (Sigma). Tissue cryosections with thicknesses of 5–20 μm were adhered to the coverslips and kept at −20 °C until used. For DNA-PAINT experiments, the tissues were labeled with mouse primary anti ryanodine or anti actinin antibodies, and targeted by the oligonucleotide conjugated secondary antibodies. Immunohistochemistry was performed in imaging chambers as described above by first permeabilizing the tissue with 0.1% Triton X-100 in PBS for 10 min at room temperature (RT). The samples were blocked with 1% bovine serum albumin (BSA) for 1 h in a hydration chamber. The monoclonal mouse anti-ryanodine receptor (RyR, MA3-916, Thermo Fisher) primary antibody was incubated overnight (4 °C) with the sample at 5 μg mL⁻¹ in a PBS incubation solution buffer containing 1% BSA, 0.05% Triton X-100 and 0.05% sodium azide, alpha-actinin (A7732, Sigma) was diluted 1:200 in incubation buffer and treated in the same manner. Samples were washed in PBS 3–4 times for 10–15 min each. Secondary antibodies, previously conjugated to oligonucleotides and stored at 1 mg ml⁻¹ were diluted 1:200 in incubation solution, added to the samples, and left for 2 h at RT. The tissue was then finally washed a further 3 times in PB.

*Imaging setup and analysis.* A modified Nikon Eclipse Ti-E inverted microscope (Nikon, Japan) with ×60 1.49NA APO oil immersion TIRF objective (Nikon, Japan) was used to acquire super-resolution data. Images were taken using an Andor Zyla 4.2 sCMOS camera (Andor, UK) using a camera integration time of 100 ms, or 10 ms for accelerated acquisition (Fig. 6b and Supplementary Fig. 8). A tunable LED-light source (CoolLED, UK) was used where possible to illuminate the widefield fluorescence and check labeling quality prior to super-resolution imaging. A 642 nm continuous wave diode laser (Omikron LuxX, Germany) was used to excite the ATTO 655 imager strands for DNA-PAINT imaging. Microspheres and DNA-origami tiles were imaged in total internal reflection fluorescence (TIRF) mode, whilst tissue samples required highly inclined and laminated optical sheet (HILO) mode. An auxiliary camera (DCC3240N, Thorlabs) was used in a feedback loop to monitor and correct for focal drift, similar to McGorty et al.[25], and previously implemented in ref.[6]. Red fluorescent beads with a diameter of 200 nm (F8887, ThermoFisher Scientific) were introduced to the samples prior to DNA-PAINT imaging and later used in post-analysis to correct for lateral drift.

Operation of the microscope components, image acquisition and image analysis were conducted using the Python software package PyME[26] (Python Microscopy Environment), which is available at https://github.com/python-microscopy/python-microscopy. Single molecule events were detected and fitted to a 2D Gaussian model. Localization events were rendered into raster images that were saved as tagged image file format (TIFF) either by generating a jittered triangulation of events or by Gaussian rendering[27].

*DNA-PAINT experiments.* A step-by-step protocol describing the procedure for conducting Repeat DNA-PAINT can be found at Protocol Exchange[28]. All DNA-PAINT experiments were conducted with solutions made up in PB, described above, and imaged at 10 frames/s (100 ms integration time) unless otherwise stated. Typically, the imager concentration in experiments with *n*-times docking motifs were diluted *n*-times in comparison to the concentrations used for a single docking motif on the same sample. 3′ ATTO 655 modified imagers were diluted to 0.04–0.4 nM (biological sample) and 0.2–2 nM (origami) depending on x RD present, experiment and sample in use. For experiments where 1x RD and 10x RD motifs had to be connected to anchor strands, these were added at 100 nM (biological samples) or 200 nM (origami). The azide modified anchor strand used for experiments involving biological samples was labeled with 3′ Cy5 or Cy3 fluorophore to aid with both the click-chemistry conjugation and for easily identifying a suitable location to image within the biological sample. The widefield dye was rapidly photobleached prior to DNA-PAINT imaging and therefore did not contribute to the super-resolution data. In order to switch between 1x RD and 10x RD as highlighted in Fig. 1c, the displacer strand D was introduced at ~100 nM and allowed to remove the incumbent docking motif. Washing, in order to remove excess D and D-1x RD (or D-10x RD) complexes, was conducted with the *n*-times lower imager concentration before subsequently adding the new *n*-times repeat docking motif as above. Figure 3b was rendered by jittered triangulation utilizing >40k frames for 1x RD segments.

*Microsphere test samples: event-rate quantification.* To quantify event rates in Fig. 1b microspheres decorated with 1x RD, 3x RD, or 6x RD were imaged with $[I]$ = 50 pM collecting 5000 frames. The three populations of microspheres were imaged individually ($n$ = 82, 88, 68 for 1x/3x/6x functionalized microspheres) in a split imaging chamber but using the same imager solution to guarantee an equal imager concentration. Event rates were calculated as mean value of the number of detected binding events per second and per individual microsphere.

*Biological tissue: event-rate quantification.* Event-rate traces in Supplementary Fig. 2 were obtained using tissue samples immuno-labeled to show the RyR with the anchor strand initially harboring 1x RD prior to being displaced and exchanged, as described above, with 10x RD. An imager concentration of 0.4 nM was used for 1x RD, while $[I]$ = 40 pM was used for the washing stage between the

removal of excess 10x RD and its imaging. The number of localized events were counted per second, by taking the sum of events collected over 10 frames (the camera integration time was set 100 ms). The entire experiment involved more than 110k frames (>3 h).

*Biological tissue: non-specific event determination.* Immunohistostained tissue with non-functionalized anchor strands only affixed to RyR, Fig. 3c, were first imaged with 40 pM P1 ATTO 655 imager (no designated complementary docking site available) and subsequently 0.4 nM in order to ascertain the level of non-specific binding. We verified that s the P1 and anchor sequences were completely non-complementary, (2) the spatial pattern that was formed by the detected non-specific events had a random appearance and bore no relationship with the specific pattern observed when docking strands were attached to anchors and (3) the temporal pattern of attachments was typical for that observed for suspected non-specific events (see also Fig. 3b). These same regions were then functionalized with 1x RD and later 10x RD docking strands and imaged again with their respective equivalent imager concentrations (1x RD [0.4 nM], 10x RD [40 pM]) as used previously. The number of events per 5 min window, repeated over a duration of 20 min, was recorded for each segment.

*Biological tissue: background measurements.* Background measurements were recorded in tissue where no imager had previously been present by measuring the mean background per 1k frames over 5k total. The intrinsic (no-imager) signal obtained was subtracted from subsequent measurements. Non-functionalized (anchor only) recordings were ascertained using the events from 5k frames for both 40 pM ($n = 712$) and 0.4 nM ($n = 5002$). When functionalized with either 1x or 10x RD the background measurement for the relative imager concentrations were obtained from events over 30k frames for each modality ($n = 683$k (1x RD), 537k (10x RD)).

*Biological tissue: fourier ring correlation maps.* Fourier Ring Correlation (FRC) measurements were performed using a PYME implementation available through the PYME-extra set of plugins (https://github.com/csoeller/PYME-extra/)[29]. After drift correction was applied the series was split into two equal blocks of events. All events were split into alternating segments containing 100 frames and these in turn were then used to generate two rendered Gaussian images which were compared using the FRC approach as described in ref. [20]. Briefly, the intersection of the FRC curve with the 1/7 line was used to obtain an estimate of the FRC resolution. In order to generate the FRC map, presented in Fig. 2c, optically thick ~20 μm porcine tissue, labeled for alpha actinin, was imaged near the surface furthest from the objective with the excitation laser orientated to pass straight out of the objective lens. For 1x RD measurements the detection threshold in the PYME analysis pipeline[26] (https://github.com/python-microscopy/python-microscopy) was set to 1.3. Because this threshold is signal-to-noise ratio based it was adjusted to 2.0 for 10x RD measurements in order to have equivalent foreground mean photon yields in detected events which ensured that equivalent detection settings are used. $2 \times 2$ μm regions of interest were individually segmented in time, utilizing 30k frames for each modality (1x/10x RD), and two Gaussian images rendered with a pixel size of 5 nm were rendered for each square. Localization precision, as shown in Fig. 2d inset, was determined by the PyME localization algorithm which estimates the localization error from the co-variance of the weighted least squares penalty function at convergence, see also[30].

*Biological tissue: accelerated sampling.* For the data summarized in Fig. 6b we initially sampled anchor strands directly as per normal DNA-PAINT experimentation, using a 9 nt P5 imager (see Supplementary Table 1), [I] = ~0.3 nM and a camera integration time of 100 ms each. Following this sequence, 10x RD was introduced at 100 nM and allowed to hybridize to the anchor. Excess 10x RD was washed out with PB. The camera integration time was decreased to 10 ms and the excitation laser intensity was also increased by removing an ND0.5 filter. A shorter P1s imager strand (8 nt) was then added at [I] = ~0.3 nM, and blinking events recorded. The total number of frames acquired was 20k in the first experimental phase and 160k in the second. FRC measurements were taken from four regions across the sample at intervals of 1k or 10k frame to obtain the plot in Supplementary Fig. 8.

*Biological tissue: widefield functionality using repeat domains.* Cardiac tissue labeled for alpha actinin were first imaged, in widefield-mode, using the Cy3 dye attached to the anchor strand, Supplementary Fig. 9. Next, the anchor strands were functionalized with 10x RD motifs and imaged in widefield-mode using a nominally low P1 ATTO 655 imager concentrations of ~1 nM, illuminated with 647 nm laser excitation and imaged with 500 ms camera integration time. After acquiring widefield data the imager concentration was reduced with a series of washes in DNA-PAINT buffer and replaced with 40 pM P1 ATTO 655 imager and imaged as normal for super-resolution.

*Origami test samples: event-rate quantification.* To quantify event rates in Fig. 1d origami tiles were first functionalized and imaged with 1x RD motifs using 2 nM P1 ATTO 655 imager. After approximately 40k frames 1x RD were displaced and

replaced with 10x RD and the imager concentration reduced by a factor of ten. Tiles identified as having had all sites occupied ($n = 49$ 1x RD and $n = 81$ 10x RD tiles) within the imaging period were used to ascertain number of events per second per tile.

*Origami test samples: resolution measurements.* Imaging resolution was assessed in origami test samples with the design in Fig. 5c, featuring a row of three point-like binding sites labeled with 1x RD or 10x RD docking domains (attached via anchor overhangs). Resolution was quantified from the intensity profiles measured across the three sites in the rendered images (Fig. 5d and Supplementary Fig. 7). Estimations of the full width at half maximum of the peaks were sampled over 30 individual sites (10 origami) for both 1x RD and 10x RD.

*Origami test samples: FRC measurements.* Single origami tiles were selected and rendered at 0.5 nm pixel size in ~210 nm² boxes and the FRC analysis, described previously in 'Biological tissue: Fourier ring correlation maps', was applied to tiles from 1x RD ($n = 47$ tiles) data series and 10x RD ($n = 80$ tiles) docking motif data series, respectively.

*Origami test samples: quantification of photoinduced site loss.* Origami tiles with 6 binding sites, with 1x RD or functionalized with 10x RD, were imaged for 40 K frames. Tiles that could be identified were then constrained to the first 20 K frames (total of 442 tiles for 1x RD origami and 285 tiles for 10x RD origami). The same tiles were then inspected in an image rendered from frame numbers 20 K to 40 K and the number of detectable sites counted again. Site loss, expressed as a percentage (Fig. 4), was specified as the difference between the sites detected in the first 20 K frames and the sites detected in the second 20 K frames.

*Origami test samples: qPAINT analysis of 6 and 5-spot tiles.* To establish compatibility of qPAINT analysis with 10x RD motifs, origami tiles as shown in Fig. 6a, with 6 and 5 spots, respectively, were selected for qPAINT analysis in the python-microscopy environment. The qPAINT analysis approach essentially follows Jungmann et al.[3]. Event time traces obtained by analysis in the PYME software environment were used to determine dark times, i.e., time intervals between detected fluorescence events. Due to dye blinking and event detection noise (e.g., events being above detection threshold in one frame but below detection threshold in a consecutive one) there was an additional distribution of very short dark times, typically <10 frames. In a cumulative histogram we modeled this behavior as resulting in a cumulative distribution function (CDF) of the form:

$$\text{CDF(t)} = \alpha\left(1 - e^{-\frac{t}{\tau_B}}\right) + (1 - \alpha)\left(1 - e^{-\frac{t}{\tau_D}}\right), \tag{2}$$

where $0 < \alpha < 1$ and the fast blinking time $\tau_B$ was constrained to be <8 frames. The dark time $\tau_D$ obtained by fitting this CDF to experimental dark time distributions was used to conduct qPAINT analysis. To calculate the number of binding sites uncalibrated qPAINT indices were determined as the inverse of dark times[6,31]. The qPAINT indices were pooled for 6 and 5 spot containing tiles, respectively. The histogram of qPAINT indices for 6-spot tiles was fit with a Gaussian as shown in Fig. 6a. The center of the fitted Gaussian was used to obtain a qPAINT index calibration value for 6–10x RD docking motifs. The calibration was applied to all data, and the qPAINT estimate of the number of 10x RD motifs on 5-spot tiles obtained through gaussian fitting of the calibrated qPAINT histogram in Fig. 6a.

## Simulation methods
*Spatial fluorophore distribution in binding events.* Estimates of the probability distributions of fluorophore locations in Fig. 5a were acquired through molecular simulations using the coarse-grained model oxDNA[15]. oxDNA is top–down parametrized and describes each nucleotide as a site with 6 anisotropic interactions: excluded volume, stacking, cross-stacking, hydrogen bonding, backbone connectivity and electrostatic repulsion. Here we used the updated oxDNA2 force field with explicit electrostatics[32].

The systems were simulated using Monte–Carlo (MC) sampling, and moves were proposed with the Virtual Move Monte Carlo (VMMC)[33] scheme to better sample the highly correlated degrees of freedom. The maximum VMMC cluster-size was set to 12 nucleotides, with translational moves of 0.05 oxDNA units, and rotational moves of 0.22 oxDNA units. Temperature was set to 300 K. We run simulations at effective monovalent salt concentrations of 640 mM.

Separate simulations were initialized with the imager bound to each of the possible locations on docking strands 1x RD, 3x RD, and 6x RD. Large artificial biases were used to ensure that at least 7 of the 9 imaging-docking bonds were formed, so that the two strands remained bonded for the duration of the simulation. The end-nucleotide of the docking motif corresponding to its anchoring point, was confined to point with a 3D harmonic potential.

Each system was simulated in 16 replicas, for between $9 \times 10^5$ and $2.7 \times 10^6$ MC steps. The position of the fluorophore-bearing nucleotide on the imager was taken as a proxy for that of the fluorophore (which cannot be simulated in oxDNA), and its location relative to the harmonic trap anchoring the docking motif was sampled every 500 steps. The fluorophore location was then projected onto the x-y, plane to produce the 2D probability distributions in Supplementary Fig. 6, with uncertainties calculated between replicas (which however are negligible and

unnoticeable in Fig. 5a). The probability distributions in Fig. 5a are obtained by radial averaging.

In Supplementary Note 2 we show that the timescales of relaxation of the imager-docking configuration into equilibrium are orders of magnitude faster than those of photon emission. One can thus assume that the physical locations from which photons are emitted are randomly drawn from the distributions of dye locations. The photon spatial distribution sampled by the microscope during each blink can therefore be estimated by convolving of the distribution of fluorophore locations with the PSF, here approximated with an Airy disk whose full width half maximum (FWHM) is 250 nm. Convolution between the PSF and fluorophore distributions is performed in 2D, and the radial cross sections are shown in Fig. 5b. This approximate PSF is justified as the FWHM of an Airy disk occurs at $0.51\lambda/NA \approx 250$ nm, using values of $\lambda = 700$ nm and $NA = 1.45$ that closely correspond to the experimental conditions in this study.

*Evaluation of hybridization rate using forward flux sampling.* We use molecular dynamics (MD) simulations performed with the oxDNA model to estimate the relative rates of hybridization of imagers to docking motifs with variable number of repeats (1x RD, 3x RD, and 6x RD) as shown in Fig. 1b. The absolute rates are not accessible, since diffusion rates in the coarse-grained representation oxDNA are not necessarily realistic.

For these simulations, the oxDNA force field is manually modified to eliminate intra-strand hydrogen bonding. Such a modification is necessary to prevent the appearance of a hairpin loop in 6x RD. Said loop is predicted not to occur by standard Nearest-Neighbor nucleic acid thermodynamics, as implemented in NUPACK[34]. We suspect the loop formation in oxDNA is an artifact related to identical excluded volume for purines and pyrimidines, so that duplex destabilization due to base pair mismatch is underestimated.

Our objective is to estimate the first order rate constant of imager hybridization to any binding domain of a tethered docking strand. Even with the highly coarse-grained oxDNA model, hybridizations are still rare over simulated timescales. To enhance sampling of hybridization events, we use Direct Forward Flux Sampling (FFS)[35,36]. FFS relies on defining a reaction coordinate onto which the state of the system can be projected. Along this coordinate one then identifies a number of intermediate system configurations between the initial and final states of interest. The rate for the system to evolve between the initial and final states can then be decomposed over the intermediate steps, which can be sampled more effectively.

Our implementation of FFS is based on that of Ouldridge et al.[14]. We define a reaction coordinate $Q$ which can take all integer values between $Q = -2$ and $Q = 4$. For $Q = -2, -1, 0$ the reaction coordinate is defined based on to the minimum distance $d_{\min}$ between the imager and the docking motifs, calculated considering any of the nucleotides on either strand. This includes nucleotide pairs that are not-complementary. For $Q = 1...4$, the coordinate is also dependent on $N_{\mathrm{bonds}}$, the number of nucleotide bonds between docking strand and imager. Following ref. [37] we assume that two nucleotides are bound if their energy of hydrogen bonding is more negative than 0.1 simulation units, equivalent to 2.5 kJ mol$^{-1}$. $Q = 4$ corresponds to our target state in which all 9 imager nucleotides are hybridized to the docking strand. Conditions associated to all values of $Q$ are summarized in Supplementary Table 2. We indicate as $\lambda_i^{i+1}$ the non-intersecting interfaces between states with consecutive values of the reaction coordinate, where $i = -2...n - 1$. E.g. $\lambda_0^1$ is the interface between states with $Q = 0$ and those with $Q = 1$. Note that for the system to transition from $Q = -2$ to $Q = 4$ it is necessary that all intermediated values of the reaction coordinate are visited.

The rate of imager-docking hybridization can then be calculated as

$$r = \Phi_{-2\to 0} \prod_{i=1}^{4} p(i|i-1) \quad (3)$$

Here, $\Phi_{-2\to 0}$ is the flux from interface $\lambda_{-2}^{-1}$ to $\lambda_{-1}^0$, and $p(i|i-1)$ are the probabilities that when at interface $\lambda_{i-2}^{i-1}$, the system crosses interface $\lambda_{i-1}^i$ before reverting back to interface $\lambda_{-2}^{-1}$.

The flux $\Phi_{-2\to 0}$ is estimated from a simulation run as $\Phi_{-2\to 0} = \frac{N_{-2\to 0}}{T_{\mathrm{sampling}}}$, where $N_{-2\to 0}$ is the number of successful transitions from states with $Q = -2$ to states $Q = 0$ observed after simulating the system for $T_{\mathrm{sampling}}$ time steps. A successful transition is recorded every time the system first visits a state with $Q = 0$ after having occupied one with $= -2$. Prior to beginning to sample transitions, the system is equilibrated for $10^6$ time steps. Note that generating $\Phi_{-2\to 0}$ at experimentally relevant (low nM) imager concentrations would be inefficient. Instead, we place one imager and one docking strand in a cubic (periodic) box of side length 42.5 nm corresponding to an effective concentration of 21.6 μM. Time spent in hydrogen bonded states is not included in $T_{\mathrm{sampling}}$.

Subsequently, we evaluate the crossing probabilities of individual interfaces $p(i|i-1)$. We start by randomly choosing saved trajectories at $\lambda_{-1}^0$ and simulating until we either reach $\lambda_0^1$ (success) or $\lambda_{-2}^{-1}$ (failure), then record the probability of success, $p(1|0)$, as well as the instantaneous configuration on passing through $\lambda_0^1$. Then, we randomly choose from those saved trajectories at $\lambda_0^1$, and simulate until either at $\lambda_1^2$ (success) or $\lambda_{-2}^{-1}$ (failure), saving trajectories at $\lambda_1^2$, as well as the success probability $p(2|1)$. We continue this procedure for the subsequent interfaces $\lambda_2^3$ and $\lambda_3^4$, and finally obtain the imager-docking hybridization rate in Eq. 3.

Details for the number of trials and successful transitions across each interface are summarized in Supplementary Tables 3 and 4.

The on-rates in Fig. 5a are averaged between two simulation repeats of approximately 20,000 transitions through each interface.

The relative hybridization rates of imager strands to each individual binding site on the multi-repeat docking motifs, shown in Supplementary Fig. 1, are extracted from the distribution of terminal states in FFS. Note that the terminal state $Q = 4$ in our reaction coordinate is defined as one in which 9 nucleotide bonds are formed between the imager and docking strand, regardless of which nucleotides are hybridized (in Supplementary Table 2). To determine which one of the binding sites is occupied in a given FFS terminal configuration we therefore analyzed the secondary structure of the terminal configurations. We defined the imager as being bound to a given domain if the majority of the docking nucleotides participating in bonding belonged to that domain. Approximately 20,000 terminal secondary structures were analyzed for the two separate simulation runs.

Concerning precise parameters needed to replicate these simulations: MD timestep were set to 0.003 oxDNA time units (9.1 femtoseconds) with an oxDNA diffusion coefficient set to 1.25 oxDNA units. Major-minor grooving was turned off. Temperature was set to 300 K and the standard oxDNA thermostat used and set to thermalize a fraction of velocities every 51 timesteps.

**Reporting summary**. Further information on research design is available in the Nature Research Reporting Summary linked to this article.

## Data availability
All data supporting this study are available upon reasonable request. Source data are provided with this paper.

## Code availability
Experimental data was collected using the Python software package PyME (Python Microscopy Environment), which is available at https://github.com/python-microscopy/python-microscopy.[26] Simulations were carried out using a modified version of the oxDNA package available at https://github.com/WillTKaufhold1/oxDNA-no-self-bonds.[38] Data was analyzed using the Python software package PyME and the set of plugins available at https://github.com/csoeller/PyME-extra.[29]

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

## Acknowledgements

We thank Dr. Ruisheng Lin for his assistance with PyME software control. This work was supported by the Engineering and Physical Sciences Research Council of the UK (No. EP/N008235/1) and Biotechnology and Biological Sciences Research Council Grants BB/P026508/1 and BB/T007176/1. LDM acknowledges support from a Royal Society University Research Fellowship (UF160152) and from the European Research Council (ERC) under the Horizon 2020 Research and Innovation Programme (ERC-STG No 851667—NANOCELL). W.T.K. acknowledges funding from an EPSRC DTP student-ship. This work was performed using resources provided by CSD3 operated by the University of Cambridge Research Computing Service (www.csd3.cam.ac.uk), provided by Dell EMC and Intel using Tier-2 funding from the EPSRC (capital grant EP/P020259/1), and DiRAC funding from STFC (www.dirac.ac.uk).

## Author contributions

C.S. and L.D.M. supervised the research progress. Data acquisition was performed by A.H.C., T.L., W.T.K., A.M. and data analysis was performed with contributions from all authors. T.L. prepared and imaged microspheres, W.T.K. made the origami tiles, A.H.C. and T.L. prepared and imaged the origami tiles, A.H.C. and A.M. prepared and imaged biological samples. Simulations and modeling were developed by W.T.K. and L.D.M. The manuscript was written by A.H.C., W.T.K., L.D.M., and C.S. and all authors reviewed the manuscript.

## Competing interests

The authors declare no competing interests.
