## [Peer Review File · Nature Communications]

Reviewers' Comments:

Reviewer #1:

Remarks to the Author:

The manuscript by Alexander Clowsley et al., 'Repeat DNA-PAINT suppresses background and non-specific signals in optical nanoscopy' describes the development of a new strategy for DNA-PAINT 'Repeat DNA-PAINT' based on repeated 'docking' DNA motifs. The use of identical Repeated DNA sequences ($n=3, 6, \text{ or } 10$) as a docking strand, instead of one single DNA sequence, enables to perform DNA-PAINT super-resolution microscopy using decreased imager strand concentrations (e.g. from 0.4 nM to 40 pM). This results in a reduced background fluorescence (associated free imagers in the imaging solution), reduced contribution of non-specific labelling, while improving slightly the spatial resolution and preserving quantitative aspects of DNA-PAINT (qPAINT).

The authors nicely showed that reducing the imager concentration during DNA-PAINT acquisitions: i) decreased background fluorescence which enhanced slightly the pointing accuracy (Fig. 1, Supplementary Fig. 3); ii) decreased non-specific labelling (Fig. 1, Supplementary Fig. 4,5); reduced photoinduced docking site loss (Fig. 1). As pointed out by the authors, this method should greatly enhance the DNA-PAINT imaging conditions, especially when the sample is away from the glass cover-slip and could not be imaged in the TIRF mode. Experimentally the manuscript is very solid. Development of this new method will be very valuable to the cell biology community. I am rather supportive for its eventual acceptance in Nature Communications. Nevertheless, I feel that further demonstrations and illustrations of the abovementioned advantages should be added to strengthen and improve the manuscript.

Main concerns that should be addressed:

1. The evidence of an improved spatial resolution using Repeat DNA-PAINT compared to conventional DNA-PAINT are not completely exploited in the manuscript. Although, the authors showed in Supplementary Fig. 3 an improved localization precision (10.5 nm obtained with conventional DNA-PAINT (400 pM) compared to 8.7 nm with Repeat DNA-PAINT (40 pM)) this improvement is real but rather small. The authors should image samples in which the spatial resolution of conventional DNA-PAINT will be greatly compromised because of background fluorescence, and show that in those conditions Repeat DNA-PAINT will significantly improve the localization precision and thus spatial resolution.
2. Concerning reduction of non-specific labelling. Figure 1h illustrates the decreased non-specific labelling that should be reduced 10 fold according to the authors (caption of Supplementary Fig. 5 "...reduces non-specific localizations ~ 10 -fold while maintaining specific localizations at the same level"). The curves in Fig. 1g and Supplementary Fig. 4 show this reduction in non-specific labelling. Nevertheless, the authors should present in comparison two acquisitions performed at 0.4 nM and 40 pM that really illustrate this dramatic decrease in the non-specific labelling. This is one of the major strength of Repeat DNA-PAINT.

Minor comments:

1. p. 4. Could the authors specify in the manuscript, when they demonstrate the use of Repeat DNA-PAINT to increase the acquisition rate (paragraph p. 4, Fig. 2g), that performing DNA-PAINT acquisitions with 10 ms integration time versus 100 ms considerably impairs the spatial resolution.
2. p.1. Could the authors be more precise when they mentioned "in real-life biological scenarios".
3. p.2. Could the authors change 'accounting' by 'subtracting' and modify the sentence accordingly: "When accounting for the intrinsic (imager-free) signal ($11.2 \pm 0.3 \text{ photons/pixel}$), the average imagerinduced background decreased from $30.5 \pm 0.05 \text{ photons/pixel}$ at $[I] = 0.4 \text{ nM}$ to $6.3 \pm 0.04 \text{ photons/pixel}$ at $[I] = 0.04 \text{ nM}$, a ~ 5 -fold reduction."

Reviewer #2:

Remarks to the Author:

This manuscript reports an advance on the DNA PAINT method for super resolution single-molecule localisation microscopy.

Conventional DNA PAINT measures the transient binding of otherwise freely diffusing single fluorescent DNA imager strands with complementary DNA docking strands, which are tethered to the target site. Acquisition of a well-sampled distribution of binding events is required for accurate spatial localisation. DNA PAINT thus requires that imaging is performed in sufficiently high concentration of imager strands to achieve a practical binding rate that minimizes total acquisition times and subsequent photodamage to docking strands. At the same time, the free imaging strands result in background fluorescence that decreases signal to noise ratios and so cannot be too high.

The manuscript presents a conceptually simple solution – to use docking strands with multiple identical binding sites, which the authors demonstrate using oxDNA simulations, linearly increases binding rates. They also demonstrate experimentally that similar binding rates can be achieved with 10-fold lower concentration of imager strands when docking strands with 10 identical binding sites are used compared to those with single binding sites. Thus, allowing data acquisition at the same rate but with substantially lower background fluorescence. Or in principle, binding at faster rates to minimise total acquisition times.

Docking strands with multiple sites are necessarily longer than those with single binding sites and subsequently a larger radial distance distribution from their anchor point. The authors also demonstrate how this does not necessarily translate to a lower spatial localisation precision.

The manuscript thus presents a novel and useful advance on the DNA PAINT method that is of interest to the field of molecular imaging. I do however have some major concerns regarding the clarity and integrity of some data that I believe should be addressed in a published manuscript:

1. Binding event rates measured on microspheres reported in figure 1b.

It is unclear to me what the supporting data in figure 1b represents. I presume that this is the binding rate per single docking strand? But it appears from the inset in Figure 1b that the microbeads were coated in docking strands (NB: the figure legend states that the inset is a rendered image but it appears to be experimentally derived?). How then were the binding events normalised to a single docking strand? How many docking strands were visualised per microbead and how was this measured?

2. Comparison of binding event rates with 1xRD and 10xRD docking strands at 0.4 nM and 0.04 nM imaging strands respectively (Fig 1d.)

Presumably data represents counts over the entire field of view of a HILO image in cells? (it is not clear from the figure legend). This seems to be a rather crude approach to demonstrate binding rates. This is because non-specific binding events will also be incorporated into this data and given that ryanodine receptors tend to cluster, it may be difficult to determine the number of docking strands at any location?

Rather than summing all binding events in the field of view, binding frequency should be reported per docking strand with non-specific events removed from the dataset. This would normalise for unexpected differences between the 1xRD and 10xRD, for example in case the 10xRD happened to more densely populate the sample. Better still, I suggest also presenting these data from DNA origami tile samples, where each docking strand can be unambiguously identified. Presumably the authors have these data on hand given that they also present super resolution images of docking sites on DNA origami tiles.

3. I do not believe that the conclusion that there is greater % site loss for docking strands with 1 compared with 10 identical binding sites is well supported by the data presented in its current form (Fig 1i).

Neither the example images presented of a few single particles nor the bar chart, which does not contain error bars (currently incorrectly referred to as a histogram in the figure legend) provide a sufficient view of the underlying data. If the conclusion is well supported, it should be straightforward to present the underlying data more thoroughly. For example, a histogram of the number of spots per tile between frames 0-20k and 20-40k could be presented along with the mean number of spots and corresponding variants.

4. The claim that non-specific binding events "are far (10 times) less common when targeting 10xRD.." is not substantiated.

The authors refer to figure 1h panel iii, which as far as I understand, compares the frequency of binding events at two positions not localised to a docking strand in a sample with 1xRD docking sites, to the frequency at a site consisting of a 10xRD docking strand. I do not see how these data portray differences in non-specific binding events.

5. The authors claim the method can reduce acquisition times because multiple binding sites increases the rate of binding events. However, this has not been demonstrated and it is not clear to what extent acquisition times can be increased. For example by comparing data collected with 10xRD docking strands but at concentrations typically used in conventional DNA PAINT experiments ($\sim 0.4\text{nM}$). It is conceivable that at sufficiently high concentrations multiple fluorescent imager strands will bind simultaneously to the same docking strand.

Responses to reviewer comments

We thank the Reviewers for their insightful comments which we have addressed in the revised main text and SI, featuring additional data and refined analysis. A point-by-point response is provided below.

Please note that in response to the comments and following the introduction of new data which they triggered, we have re-structured all figures. In some cases, clarity demanded distributing content over smaller figures with fewer panels so that new figure numbers are Fig. 1 to Fig. 6. The correspondence to the relevant parts of Figs. 1 and 2 in the original manuscript should be clear from the respective context of the responses below.

For the Reviewers' perusal we also included a marked version of the main text where altered text has been highlighted in green.

Reviewer #1 (Remarks to the Author):

The manuscript by Alexander Clowsley et al., 'Repeat DNA-PAINT suppresses background and non-specific signals in optical nanoscopy' describes the development of a new strategy for DNA-PAINT 'Repeat DNA-PAINT' based on repeated 'docking' DNA motifs. The use of identical Repeated DNA sequences ($n=3, 6, \text{ or } 10$) as a docking strand, instead of one single DNA sequence, enables to perform DNA-PAINT super-resolution microscopy using decreased imager strand concentrations (e.g. from 0.4 nM to 40 pM). This results in a reduced background fluorescence (associated free imagers in the imaging solution), reduced contribution of non-specific labelling, while improving slightly the spatial resolution and preserving quantitative aspects of DNA-PAINT (qPAINT).

The authors nicely showed that reducing the imager concentration during DNA-PAINT acquisitions: i) decreased background fluorescence which enhanced slightly the pointing accuracy (Fig. 1, Supplementary Fig. 3); ii) decreased non-specific labelling (Fig. 1, Supplementary Fig. 4,5); reduced photoinduced docking site loss (Fig. 1). As pointed out by the authors, this method should greatly enhance the DNA-PAINT imaging conditions, especially when the sample is away from the glass cover-slip and could not be imaged in the TIRF mode. Experimentally the manuscript is very solid. Development of this new method will be very valuable to the cell biology community. I am rather supportive for its eventual acceptance in Nature Communications. Nevertheless, I feel that further demonstrations and illustrations of the abovementioned advantages should be added to strengthen and improve the manuscript.

1. The evidence of an improved spatial resolution using Repeat DNA-PAINT compared to conventional DNA-PAINT are not completely exploited in the manuscript. Although, the authors showed in Supplementary Fig. 3 an improved localization precision (10.5 nm obtained with conventional DNA-PAINT (400 pM) compared to 8.7 nm with Repeat DNA-PAINT (40 pM)) this improvement is real but rather small. The authors should image samples in which the spatial resolution of conventional DNA-PAINT will be greatly compromised because of background fluorescence, and show that in those conditions Repeat DNA-PAINT will significantly improve the localization precision and thus spatial resolution.

In response to this comment we have examined a relevant biological situation in which fluorescent background levels are naturally pronounced. Specifically, we imaged several microns into a $20 \text{ }\mu\text{m}$ thick cardiac tissue sample, labelled for alpha actin. In this situation, Repeat DNA-PAINT with $10\times$ RD docking strands and 10 -fold reduced imager concentration resulted in $\sim 5\times$ lower background compared to conventional DNA-PAINT ($1\times$ RD). This translated into an improvement in localisation precision from 13.7 to 8.8 nm and an improved FRC resolution from 124 nm to 78 nm . We have detailed these results in a revised Figure 2, included below. We thank the reviewer for suggesting this test which has helped us to fully demonstrate the significant effect of Repeat DNA-PAINT on resolution. The new figure (pg 3) and text accompanying it are shown in the revised manuscript on pg 3-4.

2. Concerning reduction of non-specific labelling. Figure 1h illustrates the decreased non-specific labelling that should be reduced 10 fold according to the authors (caption of Supplementary Fig. 5 “...reduces non-specific localizations ~10-fold while maintaining specific localizations at the same level”). The curves in Fig. 1g and Supplementary Fig. 4 show this reduction in non-specific labelling. Nevertheless, the authors should present in comparison two acquisitions performed at 0.4 nM and 40 pM that really illustrate this dramatic decrease in the non-specific labelling. This is one of the major strength of Repeat DNA-PAINT.

We agree that the original presentation of the data lacked clarity. We have therefore replaced the original experiment with one where we initially measure just the non-specific event rate and then compare that to the total event rate which includes both specific events (imagers attaching to docking strands) as well as the non-specific events. This data is now presented in a revised figure (see below), our new Fig. 3, specifically panel Fig. 3c. Importantly, the new approach does not rely on computationally identifying and removing non-specific events, which is sometimes impossible and the very same inconvenience that Repeat DNA-PAINT avoids. The new data shows that, in conventional DNA-PAINT data and on a typical sample, non-specific binding comprises ~8% of all events. By contrast, using repeat DNA-PAINT with 10x RD docking strands lowers this to below 1%. We believe that the new approach demonstrates the ability of Repeat DNA-PAINT to suppress non-specific binding much more clearly and robustly.

Besides the new Fig. 3c, we now extensively discuss the new content on pages 4 & 5 of the revised MS.

Minor comments:

1. p. 4. Could the authors specify in the manuscript, when they demonstrate the use of Repeat DNA-PAINT to increase the acquisition rate (paragraph p. 4, Fig. 2g), that performing DNA-PAINT acquisitions with 10 ms integration time versus 100 ms considerably impairs the spatial resolution.

We thank the reviewer for this comment and have now clarified that data acquired with 10 ms integration time do have a slightly lower limiting spatial resolution, namely ~100 nm as opposed to ~80 nm when acquired with 100 ms integration time (as we show in supplementary Fig. 7). In addition, we mention that spatial resolution can in principle be further improved when using faster camera speeds by increasing the excitation power, so that the number of photons collected from a dye molecule in a short frame equal that in a longer integration time frame. In our experiments we lacked sufficient laser power to achieve this, but there are no intrinsic technical limitations if a more powerful excitation source is available.

The relevant revised text can be found on page 7 of the manuscript.

2. p.1. Could the authors be more precise when they mentioned “in real-life biological scenarios”.

We have changed this sentence to read ‘The unparalleled flexibility of DNA-PAINT comes at a cost, in the form of a number of serious drawbacks currently limiting the applicability and performance of the technology when imaging biological cells and tissues’.

3. p.2. Could the authors change ‘accounting’ by ‘subtracting’ and modify the sentence accordingly: “When accounting for the intrinsic (imager-free) signal (11.2 ± 0.3 photons/pixel), the average imager induced background decreased from 30.5 ± 0.05 photons/pixel at $[I] = 0.4$ nM to 6.3 ± 0.04 photons/pixel at $[I] = 0.04$ nM, a ~5-fold reduction.”

We believe the reviewer means for us to change the words “accounting for” to instead read “subtracting”.

This sentence has been moved into the methods section in the revised manuscript (pg. 10). In the methods section with subheading "Biological tissue: Background measurements" we now state, as suggested by the reviewer, "The intrinsic (no-imager) signal obtained was subtracted from subsequent measurements."

Reviewer #2 (Remarks to the Author):

This manuscript reports an advance on the DNA PAINT method for super resolution single-molecule localisation microscopy.

Conventional DNA PAINT measures the transient binding of otherwise freely diffusing single fluorescent DNA imager strands with complementary DNA docking strands, which are tethered to the target site. Acquisition of a well-sampled distribution of binding events is required for accurate spatial localisation. DNA PAINT thus requires that imaging is performed in sufficiently high concentration of imager strands to achieve a practical binding rate that minimizes total acquisition times and subsequent photodamage to docking strands. At the same time, the free imaging strands result in background fluorescence that decreases signal to noise ratios and so cannot be too high.

The manuscript presents a conceptually simple solution – to use docking strands with multiple identical binding sites, which the authors demonstrate using oxDNA simulations, linearly increases binding rates. They also demonstrate experimentally that similar binding rates can be achieved with 10-fold lower concentration of imager strands when docking strands with 10 identical binding sites are used compared to those with single binding sites. Thus, allowing data acquisition at the same rate but with substantially lower background fluorescence. Or in principle, binding at faster rates to minimise total acquisition times.

Docking strands with multiple sites are necessarily longer than those with single binding sites and subsequently a larger radial distance distribution from their anchor point. The authors also demonstrate how this does not necessarily translate to a lower spatial localisation precision.

The manuscript thus presents a novel and useful advance on the DNA PAINT method that is of interest to the field of molecular imaging.

I do however have some major concerns regarding the clarity and integrity of some data that I believe should be addressed in a published manuscript:

1. Binding event rates measured on microspheres reported in figure 1b.

It is unclear to me what the supporting data in figure 1b represents. I presume that this is the binding rate per single docking strand? But it appears from the inset in Figure 1b that the microbeads were coated in docking strands (NB: the figure legend states that the inset is a rendered image but it appears to be experimentally derived?). How then were the binding events normalised to a single docking strand? How many docking strands were visualised per microbead and how was this measured?

We measure the number of events per bead per second, and we only make relative comparisons, accordingly absolute docking strand numbers are not required. While we do not know the absolute number of docking strands present on each bead, we do know that the beads, all taken from the same batch, have the same average density of biotin binding sites. It is therefore expected that, to a very good approximation, the same number of docking strands bind to biotin sites on each bead, as they were functionalised under identical conditions and with large excess of docking strands. The only difference between functionalised beads is then the number of binding domains per docking strand (1x RD, 3x RD, 6x RD) as we indicate on the graph. We can therefore claim that the detected linear increase of the *per bead* rate with increasing repeat number (and fixed imager concentration) translates into an analogous trend in the rate *per docking strand*.

To address any remaining uncertainty, we have adopted the excellent suggestion of the Reviewer, given further below, namely to test this also on DNA origami tiles where the number of docking strands is known precisely. These data, included as Fig. 1d in the revised MS (pg 2), show a similar *per site* event rate recorded with 1x RD motifs and 10x RD after a 10-fold reduction in imager concentration, quantitatively confirming the expected behaviour. We believe that the new experiment, along with the original measurements, unequivocally establishes the linear dependence of event rate on the product of repeat number and imager concentration.

2. Comparison of binding event rates with 1xRD and 10xRD docking strands at 0.4 nM and 0.04 nM imaging strands respectively (Fig 1d.)

Presumably data represents counts over the entire field of view of a HILO image in cells? (it is not clear from the figure legend). This seems to be a rather crude approach to demonstrate binding rates. This is because non-specific binding events will also be incorporated into this data and given that ryanodine receptors tend to cluster, it may be difficult to determine the number of docking strands at any location?

Rather than summing all binding events in the field of view, binding frequency should be reported per docking strand with non-specific events removed from the dataset. This would normalise for unexpected differences between the 1xRD and 10xRD, for example in case the 10xRD happened to more densely populate the sample. Better still, I suggest also presenting these data from DNA origami tile samples, where each docking strand can be unambiguously identified. Presumably the authors have these data on hand given that they also present super resolution images of docking sites on DNA origami tiles.

We thank the Reviewer for this excellent suggestion. We have carried out the recommended origami measurement, which, as discussed above shows the expected behaviour. See new Fig. 1d, Page 2 (also below).

We have removed the event rate plot from Fig. 1 as the new origami data makes the point much more clearly. We further investigate the contribution of non-specific events in the new Fig. 3 (Page 4) where we quantify the fraction of non-specific events as ~8% and 0.9% for 1x RD and 10x RD, respectively. Note that it is generally not possible to remove all non-specific event post-acquisition as there is an identification problem, which is why Repeat DNA-PAINT is particularly useful. We believe that these new data solve the issues the Reviewer remarks on above. See also our reply to point 4 below for further details.

3. I do not believe that the conclusion that there is greater % site loss for docking strands with 1 compared with 10 identical binding sites is well supported by the data presented in its current form (Fig 1i).

Neither the example images presented of a few single particles nor the bar chart, which does not contain error bars (currently incorrectly referred to as a histogram in the figure legend) provide a sufficient view of the underlying data. If the conclusion is well supported, it should be straightforward to present the underlying data more thoroughly. For example, a histogram of the number of spots per tile between frames 0-20k and 20-40k could be presented along with the mean number of spots and corresponding variants.

We thank the Reviewer for this suggestion. We have now carried out the suggested analysis as shown below, and present the statistics in the revised Fig. 4b-c (pg 5) and accompanying text. See also our response to Reviewer 1 above.

4. The claim that non-specific binding events “are far (10 times) less common when targeting 10xRD..” is not substantiated.

The authors refer to figure 1h panel iii, which as far as I understand, compares the frequency of binding events at two positions not localised to a docking strand in a sample with 1xRD docking sites, to the frequency at a site consisting of a 10xRD docking strand. I do not see how these data portray differences in non-specific binding events.

We agree that the original presentation and data was not sufficiently clear. We have therefore conducted new experiments to make this point more compelling and quantitative. The new data and analysis are shown in Fig. 3, specifically Fig. 3c. We first consider a sample lacking the docking motifs, where only non-specific events are possible, and count these events over a fixed region, number of frames, and using both high and low imager concentrations. We then apply docking motifs 1x RD and 10x RD on the same samples and regions, and again image with high (1x RD) and low (10x RD) imager concentration, thus recording both specific (i.e. imagers attaching to docking strands) and non-specific events. The data demonstrate that conventional DNA-PAINT data recorded with a 1x RD docking strand contain ~8% of non-specific events. By contrast, using repeat DNA-PAINT with 10x RD docking strands the fraction is lowered to ~0.9%.

Note that his approach does not rely on computational identification of the non-specific events, a strategy that is not applicable in many instances and that the use of Repeat DNA-PAINT renders unnecessary in view of the intrinsic suppression of non-specific events.

Note also that when performing the new tests we have always verified that the features of the biological samples are reproduced nearly identically with both 1x RD and 10x RD. This strongly suggests that there is no relative difference in access of 1x RD vs 10x RD strands to the anchor site on the markers, as would also be expected due to the compact size, flexibility and diffusibility of the single-stranded DNA making up 1x RD and 10x RD strands.

In addition, the spatial pattern that was formed by the non-specific events had a random appearance and bore no relationship with the specific pattern observed when docking strands were attached. The temporal pattern of attachments for these non-specific events was typical for that observed for non-specific events (i.e. brief and not recurring).

Besides the new Fig. 3c, we extensively discuss the new content on pages 4-5 of the revised MS.

5. The authors claim the method can reduce acquisition times because multiple binding sites increases the rate of binding events. However, this has not been demonstrated and it is not clear to what extent acquisition times can be increased. For example by comparing data collected with 10xRD docking strands but at concentrations typically used in conventional DNA PAINT experiments (~0.4nM). It is conceivable that at sufficiently high concentrations multiple fluorescent imager strands will bind simultaneously to the same docking strand.

We believe we were not sufficiently clear in this section of the original MS. We have therefore expanded it and improved the clarity of the description and the arguments.

We now explicitly state that the acceleration is based on using a shorter imager (P1s) (pg 7) that has a ~10 times shorter attachment time. This use of the shorter imager P1s for acceleration may not have been sufficiently clear in the original MS. We believe this is now greatly improved.

The rationale behind imaging acceleration is as follows:

In order to reconstruct a super-resolved image, one needs to record *roughly* a fixed number of imager-docking binding events. For imaging to occur faster, these events need to be collected faster.

To collect more events per unit time, one needs to fulfil 2 conditions:

1) Increase the frame-rate, or in other words shorten the exposure time of each frame (e.g. from 100 ms to 10 ms, as in our case). The duration of the binding events also needs to be reduced to be similar to the frame integration time, otherwise multiple frames will sample the same event, thus defying the purpose of an increased frame rate. Shorter attachment times are obtained with shorter imagers, e.g. our P1s.

2) Increase the frequency of binding events, so that a sufficiently high number of events are collected in each frame, which can be done by increasing the concentration of imagers in solution. More generally, in DNA-PAINT one would like to increase imager concentration to the point at which the number of events per frame is maximised, without resulting in spatially overlapping events.

Here is where Repeat DNA-PAINT is really beneficial: In conventional (1x RD) DNA-PAINT, if the frame duration is reduced by a factor 10 (condition 1), then imager concentration of the briefer binding imager (here P1s) needs to be increased by the same factor in order to increase binding frequency and preserve the same optimal number of events detected per frame (condition 2). However, a 10-fold increase in imager concentration also makes the free-imager background unacceptably high, making it difficult to even detect binding events. Hence, this entire strategy for increasing imaging speed becomes unsustainable.

In turn, with Repeat DNA-PAINT, specifically 10x RD docking motifs, we can fulfil condition 2 without the need of increasing imager concentration with respect to a standard 1x RD DNA-PAINT experiment, owing to the fact that imager binding frequency is already accelerated by the 10-fold increase in docking repeat number. Consequently, with Repeat DNA-PAINT one can perform fast imaging at the same imager concentrations used with 1x RD conventional DNA-PAINT.

We explain these concepts in the text accompanying what is Fig. 6b (pg 7) in the revised MS. In addition, we provide a more detailed argument in supplementary note 1.

We also refer to supplementary Fig. 7 which shows that an equivalent resolution is reached with the P1s imager and 10ms/frame in about 1/6th of the time in which that resolution is achieved with the normal P1 imager at 100s/frame.

The difference to a full 10-fold acceleration can be attributed to the fact that fewer photons are collected in a shorter P1s event compared to a longer P1 event, so in reality more events need to be collected in the fast acquisition mode compared to the traditional mode. This, however, is not an intrinsic limitation as one could simply avoid it by increasing laser excitation power. This could not be done with our setup, but could easily be achieved with a more powerful laser source.

We hope that with the additional text and reference to supplementary note 1 and supplementary Fig. 8 this is now much clearer and thank the reviewer for pointing out the potential confusion.

Finally, we note that the reviewer points out correctly that the imager concentration can in principle be increased so much that eventually multiple fluorescent imager strands may bind simultaneously to the same docking strands. In all demonstrations shown in Figs. 1-6 (i.e. Fig. 1 & 2 in the original MS) conditions were carefully chosen so that we operate in the super-resolution regime and we give details on the relevant considerations in supplementary note 1 which contains a first order theory how

imager concentrations in Repeat DNA-PAINT should be chosen to stay in the super-resolution imaging regime.

Indeed, the reviewer has raised an important point here that has prompted us to add an additional supplementary figure (Supp. Fig 8). While imager concentrations at which several imagers are bound to different docking domains on the same Nx RD strand are to be avoided for super-resolution imaging, 10x RD docking strands allow diffraction-limited imaging of the stained samples, e.g. using widefield or confocal imaging, by deliberately going into this regime. This is very useful to explore samples and find biologically interesting regions of the sample using diffraction-limited imaging which can subsequently be interrogated by super-resolution imaging, merely by reducing imager concentration suitably. The great advantage of Repeat DNA-PAINT markers is that this diffraction-limited imaging can be conducted at high contrast (i.e. the imager concentrations are still sufficiently low) and it is effectively photo-bleaching free as imagers attach and detach constantly, so that fresh imagers replace any photo-bleached imagers.

We now mention this observation in the main text and provide supplementary Fig. 8 which shows that this works well in practice, as included below.

Reviewers' Comments:

Reviewer #1:

Remarks to the Author:

The revised manuscript by Alexander Clowsley et al., 'Repeat DNA-PAINT suppresses background and non-specific signals in optical nanoscopy' is improved in several ways. The authors made a significant work to address the questions raised, the answers were satisfying.

Nevertheless, in figure 2c, it is almost impossible to see the super-resolved images behind the Fourier Ring Correlation resolution maps. I would recommend the authors to show these images with and without the Fourier Ring Correlation resolution maps, or to use a color coding directly on the detections.

At the exception of the point raised above, I am satisfied with answers made by the authors.

Reviewer #2:

Remarks to the Author:

The additional data and revised descriptions have in my opinion much improved the clarity of the manuscript and more thoroughly substantiate the conclusions.

However, I see three further (relatively minor) issues in the new data presented:

First, in Figure 2b, which shows the background intensity at two different imager strand concentrations. 2 data points are not sufficient to demonstrate a linear trend. I therefore believe that it is misleading to present the data with a linear fit and to indicate in the main text that there is a "linear increase of the fluorescent background with [I]." I suggest a different presentation. (Bar chart?) Moreover, a lower background intensity with 40 pM imaging strand compared with 400 pM imager strand is expected and unsurprising. I think it is more important to show that along with the reduction in background intensity, event rates at binding sites were relatively similar between 40 pM imager and 10x RD, and 400 pM imager and 1x RD within the same experiment. Such frequencies were shown with DNA origami experiments in Figure 1 and in a separate experiment in Figure 3.

Second, the data in Figures 2d and 2e are not adequately described. How was the mean FRC resolution calculated from the data presented in 2d? Were the distributions fit to some function [the distributions do not appear to be well sampled] or were means calculated numerically? How were the localization precision measurements in the inset calculated? Figure 2e is not described or indeed referred to in the main text.

Third, in figure 3b, which compares 'non-specific' binding rates at an 'unlabeled area', by which the authors presumably mean that there are an absence of docking strands, to a labelled area, where there are docking strands within the same field of view. It is not clear to me how the authors can determine whether there is a docking strand present or not at any particular site. I think that they need to provide evidence for the presence or absence of docking strands otherwise their conclusions on the relative specific and non-specific binding frequencies are invalid. For example, if docking strands were also fluorescently labelled co-localisation data could be presented

Responses to reviewer comments

We thank the Reviewers for their insightful comments which we have addressed in the revised main text and SI. A point-by-point response is provided below.

With the revision we also include a marked version of the main text where altered text has been highlighted in green.

Reviewer #1 (Remarks to the Author):

The revised manuscript by Alexander Clowsley et al., 'Repeat DNA-PAINT suppresses background and non-specific signals in optical nanoscopy' is improved in several ways. The authors made a significant work to address the questions raised, the answers were satisfying.

Nevertheless, in figure 2c, it is almost impossible to see the super-resolved images behind the Fourier Ring Correlation resolution maps. I would recommend the authors to show these images with and without the Fourier Ring Correlation resolution maps, or to use a color coding directly on the detections.

We thank the reviewer for their comments. As suggested we have included a supplemental figure (Supp. Fig. 3) in the revised submission which shows the FRC resolution maps and underneath the underlying super-resolution rendered images.

At the exception of the point raised above, I am satisfied with answers made by the authors.

Reviewer #2 (Remarks to the Author):

The additional data and revised descriptions have in my opinion much improved the clarity of the manuscript and more thoroughly substantiate the conclusions.

However, I see three further (relatively minor) issues in the new data presented:

First, in Figure 2b, which shows the background intensity at two different imager strand concentrations. 2 data points are not sufficient to demonstrate a linear trend. I therefore believe that it is misleading to present the data with a linear fit and to indicate in the main text that there is a "linear increase of the fluorescent background with [I]." I suggest a different presentation. (Bar chart?) Moreover, a lower background intensity with 40 pM imaging strand compared with 400 pM imager strand is expected and unsurprising. I think it is more important to show that along with the reduction in background intensity, event rates at binding sites were relatively similar between 40 pM imager and 10x RD, and 400 pM imager and 1x RD within the same experiment. Such frequencies were shown with DNA origami experiments in Figure 1 and in a separate experiment in Figure 3.

We thank the reviewer for their comments. We agree that 2 points were not sufficient to allude to a linear trend. As suggested we have now displayed the same data in the form of a bar chart.

We have added a sentence to the figure legend confirming event frequencies were similar between the two modalities.

Second, the data in Figures 2d and 2e are not adequately described. How was the mean FRC resolution calculated from the data presented in 2d? Were the distributions fit to some function [the distributions do not appear to be well sampled] or were means calculated numerically? How were the localization precision measurements in the inset calculated? Figure 2e is not described or indeed referred to in the main text.

We apologise, we did not adequately explain in the figure legend that the FRC resolution distribution in 2d were obtained from the calculated FRC measurements per segment overlaid on the resolution maps in 2c. We have now corrected this in the figure legend and added text to describe this more clearly.

Localisation precision was extracted from the PyME analysis software which estimates the localisation error from the co-variance of the weighted least squares penalty function at convergence, as described in Lin et al. 2017 which we now cite. We have added a short description to the methods section.

We thank the reviewer for pointing out we had missed reference to Figure 2e in the main text, we now refer to it within the text describing Fig. 2 on page 4.

Third, in figure 3b, which compares 'non-specific' binding rates at an 'unlabeled area', by which the authors presumably mean that there are an absence of docking strands, to a labelled area, where there are docking strands within the same field of view. It is not clear to me how the authors can determine whether there is a docking strand present or not at any particular site. I think that they need to provide evidence for the presence or absence of docking strands otherwise their conclusions on the relative specific and non-specific binding frequencies are invalid. For example, if docking strands were also fluorescently labelled co-localisation data could be presented

We agree that all methods that are used by practitioners of DNA-PAINT to computationally reject non-specific imager binding are not able to unequivocally assert if there are no docking strands present at a particular site. Rather, they compare the time signature of imager attachment with the expected time course at a docking site. Deviations from frequent attachment over the whole duration of a frame sequence are then rejected as 'non-specific imager attachments', (see e.g., Böger et al. *NeuroPhotonics* 6, 035008 (2019); Jungmann et al. *Nano Lett* 10, 4756–4761 (2010)). The justification of this approach stems from the comparison with time signatures of imager attachments in samples lacking any docking strands (e.g. the data underlying our Fig. 3a).

In general, it would be extremely difficult to unequivocally identify the presence (or absence) of a docking site at any particular location, even attempts at providing co-localisation with another docking strand marker may suffer from issues (for example, early bleaching of dyes that are used for that purpose).

Indeed, this difficulty is a major motivation for the work presented, as the reviewer is correct that the suspected non-specific attachment sequence shown in Fig. 3a could instead result from a single docking strand not revisited, for example due to stochastic damage. Therefore, reducing the probability of non-specific imager attachment by an order of magnitude in the first place (by reducing the imager concentration ~10-fold) whilst maintaining specific binding frequency, is much preferable. In that situation, identification of non-specific imager attachments is not required as the likelihood of these events is becoming insignificant as compared to direct attachments to docking strands, as we show in Fig. 3c.

To reflect the important point that the reviewer identifies, we have updated the text describing Fig. 3b, stating that these are only 'suspected' non-specific events. Importantly, the point of Fig. 3b is to illustrate the problem and we do not require the data in it for any specific conclusion.

We thank the reviewer for helping us make this point more clearly. The discussion above emphasises the importance of the alternative approach provided by Repeat DNA-PAINT, that avoids the need to identify non-specific imager attachments.